# Overview of oral health status and associated risk factors in maritime settings: An updated systematic review

Tuan P. A. Nguyen[1,2], Sanju Gautam[3]☉*, Sweta Mahato[2], Olaf Chresten Jensen[4], Arezoo Haghighian-Roudsari[5]☉*, Fereshteh Baygi[6]

1 DeGroote School of Business, McMaster University, Hamilton, Ontario, Canada, 2 Faculty of Health Science, Department of Public Health, University of Southern Denmark, Esbjerg, Denmark, 3 United Mission to Nepal, Kapilvastu, Nepal, 4 Center of Maritime Health and Society, Department of Public Health, University of Southern Denmark, Esbjerg, Denmark, 5 Faculty of Nutrition Science and Food Technology, Department of Community Nutrition, Department of Food and Nutrition Policy and Planning Research, National Nutrition and Food Technology Research Institute, Shaheed Beheshti University of Medical Sciences, Tehran, Iran, 6 Department of Public Health, Research Unit of General Practice, University of Southern Denmark, Odense, Denmark

☉ These authors contributed equally to this work.
* zenefa66@gmail.com (SG); ahaghighian@yahoo.com (AH-R)

**Data Availability Statement:** All relevant data are within the paper and its Supporting Information files

## Abstract

### Objectives

The aim of this study is to provide an updated overview of the oral health status and associated risk factors in maritime settings.

### Methods

We systematically searched PubMed, Ovid Embase, Web of Science, CINAHL and SCOPUS from January 2010 to April 2023. Two independent reviewers extracted the data. The quality of included studies was assessed using relevant assessment tools.

### Results

A total of 260 records were found in the initial search; 24 articles met the inclusion criteria. Most studies had descriptive design, and only two randomized controlled trials were found. The main oral health issues noted are oral cancer, dental caries, periodontal diseases, oral mucosal lesions, and dental emergency. Male seafarers have higher risk of oral cancers in the tongue, lips, and oral cavity while oral mucosal lesions are more prevalent among fishermen.

### Conclusions

Dental caries and periodontal diseases are prevalent in both seafarers and fishermen. The consumption of tobacco, alcohol, fermentable carbohydrate, and poor oral hygiene are risk factors that affect the oral health status at sea. The occurrence of oral diseases in maritime

**Funding:** The authors received no specific funding for this work.

**Competing interests:** The authors have declared that no competing interests exist.

setting requires more attention of researchers and authorities to develop strategies to tackle these issues.

## Trial registration

**Systematic review registration number in PROSPERO:** CRD42020168692.

## Introduction

Oral health is a key indicator that affects general health, well-being, and quality of life [1]. Good oral health enables ones perform critical functions (eating, speaking, breathing and socializing) without pain and discomfort [1]. Poor oral hygiene may lead to infection in oral cavity which is found to be risk factors of various chronic diseases [2] and also affects quality of life [3]. Poor oral hygiene is also strongly associated with oral cancer [4] which ranked as the 13th most common cancer worldwide [1]. Squamous cell carcinoma (SCC) is one of the most popular form of oral cancer [5] and many SCC are found to be developed from a precancerous condition called potentially malignant disorders [6]. Potential malignant disorders in oral cavity are often characterized by various symptoms with the most common are Leukoplakia, Erythroplakia and ulceration [7]. Actinic Cheilitis is also a common malignant disorder and precursor of SCC that is often found on lips [8].

Although oral health problems are rarely life-threatening, they still remain the major public health problems due to their high prevalence among general population [9]. According to the global burden of disease study in 2017, oral diseases such as untreated caries were reported the most prevalent non-communicable diseases (NCDs) that affected 3.5 billion people worldwide [10].

High prevalence of oral pathologies and poor oral hygiene practices are not infrequent in maritime setting [11,12]. The work at sea is often characterized by physically demanding tasks, isolation, and lack of entertainment choice, inducing unhealthy lifestyle habits (e.g. emotional eating and poor food choice) [13]. Recent studies have shown that unhealthy habits including smoking, alcohol drinking, and sweet consumption are very popular in maritime settings [14–16].

Nicotine from tobacco and fermentable carbohydrates can increase the risk of periodontal diseases and dental caries, respectively [17,18]. Smokeless tobacco is common among fishermen in some regions that contains chemical carcinogens [19,20] that may trigger abnormal transformation in mucosa lesions causing leukoplakia, oral ulceration and oral cancer [21,22]. Seafaring is known for frequent exposure to solar ultraviolet radiation, which is considered as a major risk factor in the occurrence of actinic cheilitis and lip cancer [8,23]. The less diverse food choice and rotating working hours encourage the consumption of unhealthy foods onboard [24,25]. Besides, the access to oral care onboard is very limited, as not all vessels have access to dental professionals or equipped dental facilities [26,27]. Oral hygiene at sea during long voyages has also been described as being "totally neglected", and self-reported toothbrushing habits were less frequent than ashore [27]. So, the combination of such factors (e.g. deleterious habits, nature of the workplace, and inequality in health care access) have resulted in detrimental effects on oral health status at sea. It is crucial to address oral health as a major issue that might affect the overall health and well-being of seafarers. So, there should be appropriate strategies to tackle these problems. In this regard, it is very important to have comprehensive and updated information of oral health status in maritime setting. The present study set to systematically review all available evidence related to the issue of oral health status at sea.

To the best of our knowledge, there is one systematic review about the oral health of seafarer with underrepresented data about oral health in civilian sector and less developed countries as there might not many studies conducted on this population at that time [27]. In order to improve the quality and representativeness of oral health status of seafarers, an updated systematic review is essential. Eventually, the current study may enhance the quality and balance of the existing evidence in this topic.

## Materials and methods

### Identification of relevant studies

The review protocol is registered at International Prospective Register of Systematic Reviews (PROSPERO) with registration number: CRD42020168692. This study followed the preferred reporting items for systematic review (PRISMA) 2020 updated guideline [28]. We searched the most comprehensive databases such as PubMed, Ovid Embase, CINAHL, Scopus, and Web of Science (WOS) from January 2010 to April 2023. Search strategy details for all the databases are provided in S1 Appendix.

### Inclusion and exclusion criteria

Studies are eligible if they meet the following inclusion criteria: (1) all observations studies (e.g. cross-sectional, cohort, or case studies) as well as Randomize Control Trials (RCT) studies, (2) containing at least one of the target populations either seafarers or fishermen or both, (3) studies addressing oral health as one of the primary outcomes, (4) articles published after 2010 in order to avoid duplications with the previous systematic review [27], (4) publications in English. Grey literature including technical reports or doctorate thesis were considered eligible. The exclusion criteria included (1) case reports, reviews, or systematic reviews and (2) non-peer-reviewed publications.

### Data extraction and quality assessment

The extraction and quality assessment of data were conducted by two independent reviewers, and any discrepancy between them was discussed with the senior author. Data were extracted with relevant outcome of oral health including the publication year, author information, country of publication, study design and period, sample size, participants' mean age, data collection method, and relevant outcome of oral health. The findings included oral diseases prevalence, treatment, and oral health risk factors.

The critical appraisal checklist for studies reporting the prevalence data from Joanna Briggs institute was applied to assess the quality of included observational studies [29]. The tool was validated and well-accepted [29]. The tool comprises nine items which helps to assess the methodology quality including sampling method, data collection, and data analysis [30]. Each item can be answered either yes/no or unclear/not applicable with the score ranging from 0 to the maximum score 9 [31]. Quality of studies can be rated from low, medium to high with one "yes" on each answer is equivalent to one score [32]. A study with a score from 0–3 can be classified as low quality, a score from 4–6 is medium and score above 6 can be rated as high quality [33]. The quality of the RCT was assessed using the Revised Cochrane risk of bias tool for randomized trials that evaluates five domains including risk of bias arising from the randomization process, deviations from the intended interventions (effect of assignment to intervention and effect of adhering to intervention), missing outcome data, outcome measurement, and selection of reported results [34]. Overall risk-of-bias judgement is low, some concerns and

high risk. A study is rated as high risk or some concerns if at least one domain is assessed as high risk or some concern risk of bias; or low risk when all domains are rated as low risk.

## Statistical analysis

Data synthesis was the main strategy. The heterogeneity of the included studies in terms of the study methods and outcome measurements hampered the possibility of a meta-analysis.

# Results

## Identification of relevant studies

A total of 255 papers were identified from the five databases and 5 studies were found from citation searching [35–39]. After the removal of 79 duplications and 85 studies from date filtering as only studies published after 2010 will be considered, 92 articles remained; 54 articles were removed for eligibility criteria. A total of 43 studies were retrieved for full-text screening including 38 articles from databases search and 5 studies from citation searching [35–39]. Of the 43 retrieved articles, four studies were unable to retrieve full-text [40–43], six articles were excluded because of not having relevant oral health outcomes [44–49] and nine studies were removed due to irrelevant subjects [50–58]. As a result, 24 articles met the eligibility criteria and were included in the final review (S1 Fig).

## Study characteristics

The characteristic of 24 includes studies were presented in S1 Table. The majority of studies were sixteen cross-sectional studies [11,12,20,35,37,39,59–68], five studies conducted retrospective analysis on medical records [36,38,69–71], one cohort [72] and two RCTs [73,74]. Eleven studies were done in Asian countries, with studies from India [12,20,59,60,63–67], Malaysia [62] and China [74]. Ten studies were from European countries [11,35,36,38,39,69–71,73,75] and two publications from South America (Brazil) [61,72] and one study from Africa [68]. Nineteen studies discussed oral health status and diseases of fishermen and seafarers [11,12,20,35–37,59–62,64–70,72,74] in which five studies were about military seafarers [35,37,69,73,74]. One study did not mention the studied population [39]. Two studies investigated oral cancer among seafarers and fishermen [70,71] and five studies about potential malignant disorders [12,60,61,66,72]. Four studies described the emergency related to oral diseases cases and oral injury in maritime settings [36–38,69].

## Quality assessment

In general, out of 22 included observational studies, 16 studies were rated as high quality [12,20,36,38,59,61–68,70–72], 5 studies had moderate quality [11,35,37,60,69] and only one study was rated as low quality due to not presenting data and weak methodological design [39] (S2 Appendix). When assessing the quality of RCT by the revised Cochrane risk-of-bias tool, the overall risk-of-bias judgement of both studies were to have some concern on risk of bias. One study had some concerns bias due to deviations from the intended interventions [73] while the other was from the randomization process [74]. (S3 Appendix).

## Quality synthesis

**Oral health issues.** Two studies about oral cancer of seafarers in Nordic countries revealed that people in this occupation have higher risk of developing lip, oral cavity, and tongue cancers compared to general population and other occupations [70,71]. The incidence rate of cancer in oral cavity among male seafarer are two time higher than general population [70,71] while the

possibility of developing cancer related to tongue were approximately 1.66 [71] and 1.9 times [70]. Lips cancer incidence rate was reportedly 1.76 times higher in seafarer [70].

Five studies reported the prevalence of potential malignant disorders among fishermen [12,60,61,66,72] with two studies focus on actinic cheilitis [61,72], a precancerous condition often occur on lips. The occurrence rate of abnormal oral lesions was 14.9% [12], 20.8% [60] and 30% [66]. Leukoplakia was reportedly the most common oral disorders in most studies with the occurrence rate was 67.6% [60] and 13.8% [66], followed by oral ulceration with 23.9% [60] and 7.2% [66]. The reported actinic cheilitis level at lower lip among fishermen was 11.4% [61] and 12.8% [72] with one study reported the significant relationship between actinic cheilitis and older age, skin type and sunlight exposure [61].

One study investigated the prevalence of dental caries among smoker and non-smoker fishermen with caries prevalence of 46.5% and 65.8%, respectively [63]. Five studies demonstrated high prevalence of dental caries with 82% [59], 82.6% [65], 90.9% [60], 55% [67] and periodontal diseases (99% [67]; 97.2% [60]; 85.4% [65], 100% [68] among fishermen. The prevalence of dental caries and periodontal diseases among seafarers were 88% and 75.1%, respectively [20].

One study revealed that among 9122 emergency medical records, 1.47% of the records was related to some kinds of dental problems including abscess (51.8%), decay (33.3%), and fracture (8.9%) [36]. One study about infection burden among medical events onboard cargo ships showed that among 322 illness events, 1.5% cases resulted from dental infections [38]. One paper reported that 3.69% of a German naval task group required emergency treatment during a three-month voyage [69]. The reported prevalence of dental baro-trauma among French divers was about 5.3%, leading to complications such as fracture and loss of dental restoration in 68.6% cases [75].

**Associated risk factors.** *Unhealthy lifestyle habits.* Seven studies reported the prevalence of tobacco and alcohol consumption of seafarers and fishermen [11,12,20,60,62,66,72]. Smoking cigarette was a common habit that occurred in 56.11% [11] and 25% [20] of seafarer, and 83.1% [62], 38.4% [72], 26.8% [60], 24.3% [12] and 20% [66] fishermen. The usage of smokeless tobacco was also recorded in some studies on Indian fishermen and seafarer with 62% [60] and 32% [12] using chewed tobacco, 54.9% Snuff [60] and general smokeless tobacco (14.5%, 21.9%) [20,66].

The alcohol consumption of seafarer in a daily basis was reported just less than 15% in two studies [11,20] while four studies revealed the level fishermen admitted the consumption of alcohol on vessels ranging between 15.4% and 48.8% [12,60,66,72]. One study about the consumption of dairy product or fermentable carbohydrates among seafarers showed that 55% seafarers consumed such products on a daily basis [11] and 69.4% fishermen drink coffee with sugar several times a day [62].

*Oral health hygiene.* Two studies described the oral hygiene status of fishermen with 46% and 60.6% was classified as poor oral hygiene status [59,60]. Eight studies discussed the oral health hygiene practices among seafarers [11,20] and fishermen [59,60,62,65,66,68]. One study from India reported that around 60% of seafarer used Seaweed as their common cleaning instrument [20], while more than 40% of India fishermen chewing sticks to keep the teeth clean [65,66]. In another study, 43% fishermen using toothbrush and toothpaste for oral hygiene while 32% having finger and toothpaste as their main cleaning instruments [59].

Regarding the frequency of teeth cleaning, 74% of seafarers reported daily brushing, but only 40.48% brushed their teeth twice a day [11]. Around 55% of Malaysian fishermen brushed twice a day and 37.6% cleaned their teeth once [62]. 34.9% of fishermen from Ghana cleaned their teeth twice daily [68]. One study about the quality of drinking water onboard and ashore approved the absence of fluoride in 75% of examined international ports [39].

**Dental treatment and intervention.** Two studies investigated the effect of Probiotic Lactobacillus reuteri (L. reuteri)–containing lozenges [73] and dental education plus mouthwash [74] on periodontal health of navy. The combination of dental education and mouthwash was found to have a significant effect on protecting periodontal health of seafarer during long voyage while having only dental education does not significantly improve periodontal health [74]. One RCT examined the effectiveness of probiotic consumption twice daily on improving the periodontal health of navy sailors during long deployment. Results showed that testing group experienced significant improvement on periodontal indicators of bleeding on probing (BOP), gingival index (GI), plague control record (PCR), probing attachment level (PAL) and probing pocket depth (PPD) [73] after day 14 and day 42 compared to the baseline level and placebo group.

Four studies described tooth extraction as the most common dental treatment for seafarers and fishermen [11,59,65,67]. Around 70% of fishermen in India visited dentists for extracting their teeth [65,67], while 82% of seafarers lost their teeth due to extraction [11]. In contrast, only 4.1% cases required extracting teeth on navy vessels and the most popular treatments among navy sailors were pulp treatment (50%) and dental filling (41.6%) [69]. One study in India examined the need of having prosthetic among elderly fishermen, and the findings revealed that 27.3% of the studied population required multiunit prosthesis and 23.6% full prosthesis [64]. Another study about telemedicine service at sea reported the use of medications in dental emergency cases [36]. Antibiotic was prescribed for 79% of abscess cases, 33% tooth decay, and 33% tooth fracture. The prescription of analgesics was in 88.6% abscess cases, 97% tooth decay, and 92% fracture. Mouthwash was prescribed in 40% of abscesses and 49% of tooth decay.

## Discussion

A total of 24 studies were included in this updated study. The main focus of included studies is reporting the prevalence of oral diseases, oral hygiene status, treatment, and factors affecting oral health in maritime setting. Reported oral diseases included oral cancer, dental caries, periodontal diseases, oral mucosa lesions, and dental emergency. Periodontal diseases were most prevalent in the fishermen with 85.4%, 97.2%, 99% and 100% [60,65,67,68] in four studies. The findings revealed that seafaring is a high-risk job-group for oral cancer [12,66,70–72]. Two studies listed seafarers as one of occupations that have high risk of having lip, tongue, and oral cavity cancer [70,71], while five studies reported the prevalence of oral potentially malignant disorders [12,60,61,66,72].

The possible explanation might be due to the high prevalence of tobacco (smoking and smokeless) and alcohol in this population [11,12,20,66,72] because smoking cigarette, smokeless tobacco and alcohol consumption are factors associated with the increased risk of oral malignant disorders [76,77]. This association was demonstrated in another study where the oral cancer was prevalent in 70% of old adult smokers [78]. Seafaring occupation is characterized by frequent exposure to solar radiation which is the major risk factor of actinic cheilitis and is one of the main reasons for half of lip cancer cases [79,80]. Poor nutrition with less diverse diet of healthy vegetable and fruits at sea is popular on vessels [24] which may also contribute to the increased risk of oral cancers [81].

The prevalence of oral mucosal lesion of fishermen was in line with other findings on dental outpatients in India which is 16.8% [82], but was lower compared to data from Italy on male patients above 40 (81.3%) [83]. The difference of data from these studies may result from different age group selection, different clinical examination techniques, or confounding factors that were not detected as most of these studies are cross-sectional.

Five cross-sectional studies from India used World Health Organization (WHO) oral health assessment form to clinically assess the periodontal health and dental caries status of fishermen [59,60,65,67] and seafarers [20]. Three studies from India reported the prevalence of periodontal diseases experienced by 85% [65], 97.2% [60] and 99% [67] of fishermen. Also, dental calculus was one of the major problems among 33.7% of seafarers [20]. While almost all fishermen participant in a study from Ghana with 100% having dental plague and periodontal pocket [68].

The prevalence of periodontal diseases found in this study was higher compared to other studies which reported around 50% of Indian and Ghanaian adults suffering some form of periodontal diseases [84,85]. A possible explanation for this might be a high consumption of tobacco among seafarers and fishermen due to stressful and isolated working environment [14,25] and poor oral hygiene habit in this population [68]. Moreover, long distance travel to oral care facilities and out of pocket payment are critical factors that significantly affect Ghanian population access to oral care [86].

The dental caries prevalence of fishermen was 55% [67], around 82% [59,65] and 90.9% [60], while 88% of seafarers were affected by caries [20]. The current findings also matched other findings, where the prevalence of caries status among Indian general population was 78% [87]. The high prevalence of caries might have resulted from excessive use of fermentable carbohydrate [11] in combination with ineffective oral cleaning practice of this population [11,20]. One study reported the absence of fluoride in coastal water at international ports that might also contribute to the high prevalence of caries of seafarers and fishermen [39], as fluoride is a mostly effective prevention for dental caries [88]. However, the study reported the fluoride level was rated as low quality in this systematic review [39] and more studies related to this topic should be conducted in larger scale to clarify this issue.

In the present study, caries was one of the main reasons leading to dental emergency [36] or cases related to oral diseases that required emergency care [69]. The dental emergency level was 1.48% [36] and 3.69% [69], which was also analogous with findings of other studies at sea [89,90] and onshore [91].

One intervention study examined the effect of dental education and the combination of dental education and mouthwash on periodontal health on navy seafarers [69], revealing that providing only health education does not have significant effect on improving periodontal health of navy sailors during long deployment unless it is combined with mouthwash [74]. In addition, results from a systematic review showed that dental education is effective in enhancing knowledge, attitude, and practice of oral health from school children, adolescents to teachers and mothers [92]. Such contradictory results might be basically because of different contents and approaches for instruction or duration of the education. According to the existing literature, less frequent training programs in shorter period is less effective to promote sustainable behavioral changes compared to more frequent educations [93]. Moreover, with busy and stressful working schedule, it might be also challenging for employees in maritime settings to remember and apply dental education to their daily hygiene activities. So, we think that tangible interventions such as mouthwash would be more effective to remind them of taking care of their oral health.

One study reported that the daily consumption of Probiotic L. reuteri–containing lozenges could help alleviate the deterioration of periodontal diseases among navy sailors after the period of 42 –day voyage [73]. In vitro experiment, the appearance of Lactobacillus reuteri could inhibit the growth of various bacterial periodontopathogens and oral Candida species [94,95]. The result of this study is also in line with results from other RCTs evaluating the beneficial effect of regular consumption of Probiotics L. reuteri on improving periodontal indicators and periodontal health in different populations including healthy adults [96], pregnant women [97] and chronic periodontitis patients [98]. However, most these RCTs [73,98]

conducted on small samples and further research with larger sample size in longer follow-up periods should be conducted to strengthen the protective effect of probiotics on periodontal health.

Tooth extraction was evident as the most common dental treatment of fishermen [59,65,67] and civilian seafarers [11]. The current results are in contradiction with findings on navy vessels where restorative dentistry treatment was more preferred [69]. In some navy vessels, dental facilities and dental practitioners are available that can provide instant supports to crews [69]. Hence, early symptoms of oral diseases can be detected and treated properly which can minimize the possibility of tooth extraction due to late treatments. Moreover, it also requires navy crews to qualify stringent criteria of dental classifications issued by U.S. Navy Dental corps to be deployable for missions [99]. In contrast, dental treatment is often less available on civilian vessels [27] with only some studies documented on the use of telemedicine in dental emergencies with medications as the main treatment [36]. Untreated caries or periodontal diseases without appropriate treatment procedures in long period may result in bad complications such as pain, abscess, and tooth loss [100].

The presence of caries and periodontal diseases were found as the main causes leading to tooth loss in many studies [101,102]. This helps explain the high prevalence of tooth extraction among the seafarers and fishermen in the present study where the level of caries and periodontal diseases was also very high [20,59,65,67]. The popularity of tooth extraction may also result from the poor oral behavior, lack of knowledge, and limited access to oral care in this population [103]. The current data also reveals the poor oral hygiene knowledge of seafarers [11,20] and fishermen [59,62,65,66]. The use of toothbrush for cleaning teeth was one of the less popular instruments as only used by around 20% of seafarers [20] and fishermen from India [65,66]. In other surveys in different populations, toothbrush was used as the most common oral hygiene aid [104,105]. The difference may result from the lack of adequate dental knowledge or isolated work environment that makes these oral hygiene instruments less affordable during long voyages. Seafarers brush their teeth on a regular basis, but only few brush their teeth twice a day as recommended by dental professionals [11]. This may be due to the lack of oral care knowledge or irregular shiftwork that changes normal schedules.

## Strengths and limitations of the study

The included studies in the current systematic review are mostly cross-sectional studies that makes it difficult to prove the causal relationships between risk factors and oral diseases. Moreover, studies about oral health of fishermen from developed countries are missing in this study as there were evidence only from developing countries mostly from India. Consequently, the results for oral health of fishermen are skewed towards developing countries in general and India in particular. This study has provided up-to-date evidence in both developed and developing countries. So, the results on this topic will be more representative and supportive for making preventive international policies in the future.

## Conclusion

The oral health status of seafarers and fishermen is relatively poor with high prevalence of oral diseases (e.g. oral cancer and dental caries). The finding also confirmed the popularity of unhealthy habits and low perception toward oral hygiene and seeking dental supports. As a result of data shortage on the oral health of fishermen in developed countries, conducting more studies are recommended to fulfil this gap.

## Supporting information

**S1 Checklist. PRISMA 2020 checklist.**
(DOCX)

**S1 Fig. Prisma 2020 flow chart.**
(DOCX)

**S1 Table. Data extraction form.**
(DOCX)

**S1 Appendix. Search strategy.**
(DOCX)

**S2 Appendix. Quality assessment of observational studies.**
(DOCX)

**S3 Appendix. Quality assessment of RCTs.**
(DOCX)

## Author Contributions

**Conceptualization:** Olaf Chresten Jensen, Arezoo Haghighian-Roudsari, Fereshteh Baygi.

**Data curation:** Sanju Gautam.

**Methodology:** Tuan P. A. Nguyen, Olaf Chresten Jensen.

**Project administration:** Tuan P. A. Nguyen.

**Resources:** Tuan P. A. Nguyen.

**Supervision:** Fereshteh Baygi.

**Validation:** Fereshteh Baygi.

**Visualization:** Tuan P. A. Nguyen.

**Writing – original draft:** Tuan P. A. Nguyen.

**Writing – review & editing:** Sanju Gautam, Sweta Mahato, Arezoo Haghighian-Roudsari, Fereshteh Baygi.

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
