## [Decision Letter · Decision Letter 0]

8 Nov 2022

PONE-D-22-13693Overview of Oral Health Status and Associated Risk Factors in Maritime Settings: An Updated Systematic ReviewPLOS ONE

Dear Dr. Sanju Gautam,

Thank you for submitting your manuscript to PLOS ONE. After careful consideration, we feel that it has merit but does not fully meet PLOS ONE’s publication criteria as it currently stands. Therefore, we invite you to submit a revised version of the manuscript that addresses the points raised during the review process.

I congratulate you on writing an interesting manuscript. However, I would suggest to respond to reviewers comments so as to make your manuscript more robust. 

We look forward to receiving your revised manuscript.

Kind regards,

Tanay Chaubal

Academic Editor

PLOS ONE

Journal Requirements:

NO authors have competing interests

Reviewers' comments:

Reviewer's Responses to Questions

**Comments to the Author**

1. Is the manuscript technically sound, and do the data support the conclusions?

Reviewer #1: Yes

Reviewer #2: Partly

2. Has the statistical analysis been performed appropriately and rigorously? 

Reviewer #1: Yes

Reviewer #2: N/A

3. Have the authors made all data underlying the findings in their manuscript fully available?

Reviewer #1: Yes

Reviewer #2: No

4. Is the manuscript presented in an intelligible fashion and written in standard English?

Reviewer #1: Yes

Reviewer #2: Yes

5. Review Comments to the Author

Reviewer #1: Methodology

As the date of the latest search among selected databases is Jan/22, please update the search until August/22.

Line 155-161- For the potentially malignant disorders, can you further discuss them? (eg. leukoplakia and actinic cheilitis). Is there any follow-up of these lesions?

Line 156 - Please describe the findings related to oral cancer, such as the risk of developing (percentage) compared to healthy people.

Line 160-161 - Please provide the prevalence of ulcerations and leukoplakias in percentage.

Line 170 - Five from 322 (percentage?)

Line 229 - ... dental outpatients in India (percentage?)

Line 238 - 239 - How much higher?

Line 253-254 - ... health education does (not?) affect protection of periodontal...

Figure 1 - Please use the latest version of the PRISM flow diagram (2020)

Appendix S1 - Can you provide the search strategy for the Web of Science database?

Reviewer #2: The authors present a well-conducted systematic review (SR) on a topic in which there is scarce investment in primary studies. The publications found in the search justify carrying out an update of the SR on this topic. However, the authors should consider that the previous review has important flaws, since the search performed was not comprehensive (they included just the PUBMED database) and there was no assessment of the risk of bias. This suggests that the studies included in this previous review could be critically evaluated according to the protocol conducted by the authors. Or at least, these previous results should have been better explored since the objective was to update knowledge regarding the health of maritime workers.

Some considerations:

Introduction - Review the sentence “Poor oral health not only results in systemic diseases such as cardiovascular diseases, bacterial pneumonia, and diabetes mellitus, …” we know that there is an interrelationship between oral health and general health status, but this causal relationship is still under investigation and has been presented in a simplistic way.

Introduction - In the sentence “Nicotine from tobacco and fermentable carbohydrates can increase the risk of periodontal diseases and dental caries, respectively.” Add oral cancer and potentially malignant oral disorders related to smoking (leukoplakia and erythroplakia). And address other risk factors for oral lesions such as alcoholism and sun exposure (lower lip), since these conditions were the outcome of some studies included in the review.

Introduction - In the sentence "To the best of our knowledge, there is a systematic review about the oral health of seafarer with underrepresented data about oral health in civilian sector and less developed countries." Perhaps because these primary studies did not exist at the time of this previous review...

Methods - The authors should adopt the updated version of PRISMA published in 2020. The databases cited in the methodology differ from those cited in the abstract.

Results - In the sentence "Of the 25 retrieved articles, six articles were excluded because of..." Articles excluded in phase 2 must be cited in the text or the list of the excluded references and justification must be provided in the appendices.

Results - In this sentence "In general, 13 selected scoring articles 6 to 9 were considered as having low risk of bias and acceptable quality. While five studies scored 2 to 5, which can be seen as having high risk of bias. When assessing the quality of the RCT, seven out of 11 items are fulfilled, and the quality of the RCT can be rated as moderate risk of bias.” The authors could cite the respective references as it is an important result of RS and needs to be easily accessible to the reader.

Results - In the sentence “The prevalence of smoking among seafarers was 25% and 56%, respectively.” Are these two prevalences from the studies mentioned later or is there any missing information?

Discussion: It is unclear how the authors evaluated the data from the primary studies of high risk of bias in the analysis of evidence from the frequent oral conditions in the seafarers and fisherman.

6. PLOS authors have the option to publish the peer review history of their article (what does this mean?). If published, this will include your full peer review and any attached files.

Reviewer #1: No

Reviewer #2: **Yes: **Camila de Barros Gallo

---

## [Author Response · Author response to Decision Letter 0]

14 Mar 2023

We thank you reviewer for the thoughtful inputs to our manuscript. In the revised version, we tried to address all raised points. Below, you will find point-by-point reply, including line numbers in new revised manuscripts that correspond with each raised point. 

Reviewer 1

Question: As the date of the latest search among selected databases is Jan/22, please update the search until August/22.

Answer: thank you for this suggestion, we updated the search date to August 2022. 

Question: Line 155-161- For the potentially malignant disorders, can you further discuss them? (eg. leukoplakia and actinic cheilitis). Is there any follow-up of these lesions?

Answer: for the potentially malignant disorders, we have already inputted some more details with data of Leukoplakia, ulceration and actinic cheilitis. No data about follow-up was found from included studies. (line 182)

Question: Line 156 - Please describe the findings related to oral cancer, such as the risk of developing (percentage) compared to healthy people.

Answer: we have included more detail about the risk of developing oral cancer of seafarer (standardized incidence ratio) compared to healthy population – (line 180)

Question: Line 160-161 - Please provide the prevalence of ulcerations and leukoplakias in percentage.

Answer: we have provided percentage of Leukoplakia and ulceration into the manuscript (Line 187)

Question: Line 170 - Five from 322 (percentage?)

Answer: we have switched to percentage (Line 199)

Question: Line 229 - ... dental outpatients in India (percentage?)

Answer: we have added the percentage of dental outpatients in India (Line 274)

Question: Line 238 - 239 - How much higher?

Answer: we have added the percentage of other population (Line 286)

Question: Line 253-254 - ... health education does (not?) affect protection of periodontal.

Answer: we have revised the sentence to make the meaning more explicit. (Line 309)

Question: Figure 1 - Please use the latest version of the PRISM flow diagram (2020)

Answer: we have changed to the new version of Prima glow diagram 2020. (Page 23)

Question: Appendix S1 - Can you provide the search strategy for the Web of Science database?

Answer: we have added the search strategy of Web of Science database on the Appendix 1. 

Reviewer 2

The authors present a well-conducted systematic review (SR) on a topic in which there is scarce investment in primary studies. The publications found in the search justify carrying out an update of the SR on this topic. However, the authors should consider that the previous review has important flaws, since the search performed was not comprehensive (they included just the PUBMED database) and there was no assessment of the risk of bias. This suggests that the studies included in this previous review could be critically evaluated according to the protocol conducted by the authors. Or at least, these previous results should have been better explored since the objective was to update knowledge regarding the health of maritime workers.

Question: Review the sentence “Poor oral health not only results in systemic diseases such as cardiovascular diseases, bacterial pneumonia, and diabetes mellitus, …” we know that there is an interrelationship between oral health and general health status, but this causal relationship is still under investigation and has been presented in a simplistic way.

Answer: thank you for your review, we have revised the introduction part based on your feedback (Line 55). 

Question: Introduction - In the sentence “Nicotine from tobacco and fermentable carbohydrates can increase the risk of periodontal diseases and dental caries, respectively.” Add oral cancer and potentially malignant oral disorders related to smoking (leukoplakia and erythroplakia). And address other risk factors for oral lesions such as alcoholism and sun exposure (lower lip), since these conditions were the outcome of some studies included in the review.

Answer: We have included other associated risk factors to oral cancers and potentially malignant disorders in the introduction part. (Line 79)

Question: Introduction - In the sentence "To the best of our knowledge, there is a systematic review about the oral health of seafarer with underrepresented data about oral health in civilian sector and less developed countries." Perhaps because these primary studies did not exist at the time of this previous review...

Answer: thank you for the insights, it is true that many studies about oral health of fishermen and seafarer in developing countries have published after the period of 2010. We have revised this sentence based on your comment. (Line 95) 

Question: Methods - The authors should adopt the updated version of PRISMA published in 2020. 

Answer: we have updated to the latest version of Prisma flowchart and presented in Figure 1.

Question: The databases cited in the methodology differ from those cited in the abstract.

Answer: we have changed to the database name to make it consistent with the methodology.

Question: Results - In the sentence "Of the 25 retrieved articles, six articles were excluded because of..." Articles excluded in phase 2 must be cited in the text or the list of the excluded references and justification must be provided in the appendices.

Answer: we thank the reviewer for this suggestion. We have updated the search and included in-text citations of all studies that were excluded in phase two. (Line 146)

Question: Results - In this sentence "In general, 13 selected scoring articles 6 to 9 were considered as having low risk of bias and acceptable quality. While five studies scored 2 to 5, which can be seen as having high risk of bias. When assessing the quality of the RCT, seven out of 11 items are fulfilled, and the quality of the RCT can be rated as moderate risk of bias.” The authors could cite the respective references as it is an important result of RS and needs to be easily accessible to the reader.

Answer: we have added respective references in this part. (Line 169)

Question: Results - In the sentence “The prevalence of smoking among seafarers was 25% and 56%, respectively.” Are these two prevalence from the studies mentioned later or is there any missing information?

Answer: thank you for your feedback, there was some missed data in this part, we have added more detail about prevalence of tobacco consumption (both smoking and smokeless) in this part. (Line 206)

Question: Discussion: It is unclear how the authors evaluated the data from the primary studies of high risk of bias in the analysis of evidence from the frequent oral conditions in the seafarers and fisherman.

Answer: we have evaluated the data from studies that have poor quality in this systematic review under the discussion part. (Line 298).

---

## [Decision Letter · Decision Letter 1]

6 Apr 2023

PONE-D-22-13693R1Overview of Oral Health Status and Associated Risk Factors in Maritime Settings: An Updated Systematic ReviewPLOS ONE

Dear Dr. gautam,

Thank you for submitting your manuscript to PLOS ONE. After careful consideration, we feel that it has merit but does not fully meet PLOS ONE’s publication criteria as it currently stands. Therefore, we invite you to submit a revised version of the manuscript that addresses the points raised during the review process.

We look forward to receiving your revised manuscript.

Kind regards,

Tanay Chaubal

Academic Editor

PLOS ONE

Reviewers' comments:

Reviewer's Responses to Questions

**Comments to the Author**

1. If the authors have adequately addressed your comments raised in a previous round of review and you feel that this manuscript is now acceptable for publication, you may indicate that here to bypass the “Comments to the Author” section, enter your conflict of interest statement in the “Confidential to Editor” section, and submit your "Accept" recommendation.

Reviewer #1: All comments have been addressed

Reviewer #2: (No Response)

2. Is the manuscript technically sound, and do the data support the conclusions?

Reviewer #1: Yes

Reviewer #2: Yes

3. Has the statistical analysis been performed appropriately and rigorously? 

Reviewer #1: Yes

Reviewer #2: N/A

4. Have the authors made all data underlying the findings in their manuscript fully available?

Reviewer #1: Yes

Reviewer #2: Yes

5. Is the manuscript presented in an intelligible fashion and written in standard English?

Reviewer #1: Yes

Reviewer #2: Yes

6. Review Comments to the Author

Reviewer #1: (No Response)

Reviewer #2: In general, the authors considered the suggestions and made changes that improved the quality of the manuscript. The bibliographic search of this systematic review (SR) began in 2010 with the (correct) justification of the existence of a previous SR. However, according to the PRISMA 2020 guideline, in an update of an SR, the studies included in the previous review must be also included in the updated review. This information of the PRISMA 2020 flowchart was not considered by the authors. Taking into account that the previous SR has important flaws (not comprehensive search - included just the PUBMED database – absence of risk of bias assessment), this updated SR could perform the risk of bias analysis of these previous SR, or at least report the data of previous studies with the same point of view. As it stands, the relationship with previous data from the literature was not addressed.

7. PLOS authors have the option to publish the peer review history of their article (what does this mean?). If published, this will include your full peer review and any attached files.

Reviewer #1: No

Reviewer #2: **Yes: **Camila de Barros Gallo

---

## [Author Response · Author response to Decision Letter 1]

28 Jun 2023

Rebuttal letter

We thank you reviewer for the thoughtful inputs to our manuscript. In the revised version, we tried to address all raised points. Below, you will find point-by-point reply, including line numbers in new revised manuscripts that correspond with each raised point. With extra content, the original page number may be different compared to the previous version. 

In this revision, we have updated our search until April 2023 and included five more studies that meet the scope of this systematic review. 

- N S, V., Vas, R., Uppala, H., Vas, N. V., Jalihal, S., Ankola, A. V., & K, R. S. K. (2022). Dental caries, oral hygiene status and treatment needs of fishermen and non-fishermen population in South Goa, India. International maritime health, 73(3), 125–132. https://doi.org/10.5603/IMH.2022.0025

- Tormeti, D., Nii-Aponsah, H., Sackeyfio, J., Blankson, P. K., Quartey-Papafio, N., Arthur, M., & Ndanu, T. A. (2022). Periodontal status and oral hygiene practices among adults in a peri-urban fishing community in Ghana. The Pan African medical journal, 42, 126. https://doi.org/10.11604/pamj.2022.42.126.24557

- Nithya, V. R., Krithika, C., Sridhar, C., & Arumugam, A. E. (2021). Assessment of Oral Health Care Needs among Fishermen Living in North Chennai, India – A Cross Sectional Study. Journal of Pharmaceutical Research International, 33(58B), 379–385. https://doi.org/10.9734/jpri/2021/v33i58B34214

- Schlagenhauf, U., Rehder, J., Gelbrich, G., & Jockel-Schneider, Y. (2020). Consumption of Lactobacillus reuteri-containing lozenges improves periodontal health in navy sailors at sea: A randomized controlled trial. Journal of periodontology, 91(10), 1328–1338. https://doi.org/10.1002/JPER.19-0393

- de Oliveira Ribeiro, A., da Silva, L. C., & Martins-Filho, P. R. (2014). Prevalence of and risk factors for actinic cheilitis in Brazilian fishermen and women. International journal of dermatology, 53(11), 1370–1376. https://doi.org/10.1111/ijd.12526

Reviewer 1

Question: As the date of the latest search among selected databases is Jan/22, please update the search until August/22.

Answer: thank you for this suggestion, we updated the search date to April 2023. When updating to April 2023, we have also found 5 more studies and also included in the analysis. 

Question: Line 155-161- For the potentially malignant disorders, can you further discuss them? (eg. leukoplakia and actinic cheilitis). Is there any follow-up of these lesions?

Answer: for the potentially malignant disorders, we have already inputted some more details with data of Leukoplakia, ulceration and actinic cheilitis. (line 250)

Question: Line 156 - Please describe the findings related to oral cancer, such as the risk of developing (percentage) compared to healthy people.

Answer: we have included more detail about the risk of developing oral cancer of seafarer (standardized incidence ratio) compared to healthy population. (line 206)

Question: Line 160-161 - Please provide the prevalence of ulcerations and leukoplakia in percentage.

Answer: we have provided percentage of Leukoplakia and ulceration into the manuscript (Line 250)

Question: Line 170 - Five from 322 (percentage?)

Answer: we have switched to percentage (Line 262)

Question: Line 229 - ... dental outpatients in India (percentage?)

Answer: we have added the percentage of dental outpatients in India. The prevalence of oral mucosal lesion of fishermen was in line with other findings on dental outpatients in India which is 16.8%. (Line 377)

Question: Line 238 - 239 - How much higher?

Answer: we have added the percentage of general population. The prevalence of periodontal diseases found in this study was higher compared to other studies which reported around 50% of Indian and Ghanaian adults suffering some form of periodontal diseases. (Line 390)

Question: Line 253-254 - ... health education does (not?) affect protection of periodontal.

Answer: we have revised the sentence to make the meaning more explicit. One intervention study examined the effect of dental education and the combination of dental education and mouthwash on periodontal health on navy seafarers, revealing that providing only health education does not have significant effect on improving periodontal health of navy sailors during long deployment unless it is combined with mouthwash (Line 431)

Question: Figure 1 - Please use the latest version of the PRISM flow diagram (2020)

Answer: we have changed to the new version of Prima glow diagram 2020. 

Question: Appendix S1 - Can you provide the search strategy for the Web of Science database?

Answer: we have added the search strategy of Web of Science database on the S1 Appendix. 

Reviewer 2

Feedback:

“The authors present a well-conducted systematic review (SR) on a topic in which there is scarce investment in primary studies. The publications found in the search justify carrying out an update of the SR on this topic. However, the authors should consider that the previous review has important flaws, since the search performed was not comprehensive (they included just the PUBMED database) and there was no assessment of the risk of bias. This suggests that the studies included in this previous review could be critically evaluated according to the protocol conducted by the authors. Or at least, these previous results should have been better explored since the objective was to update knowledge regarding the health of maritime workers.”

Question: Review the sentence “Poor oral health not only results in systemic diseases such as cardiovascular diseases, bacterial pneumonia, and diabetes mellitus, …” we know that there is an interrelationship between oral health and general health status, but this causal relationship is still under investigation and has been presented in a simplistic way.

Answer: thank you for your review, we have revised the introduction part based on your feedback (Line 55). 

Question: Introduction - In the sentence “Nicotine from tobacco and fermentable carbohydrates can increase the risk of periodontal diseases and dental caries, respectively.” Add oral cancer and potentially malignant oral disorders related to smoking (leukoplakia and erythroplakia). And address other risk factors for oral lesions such as alcoholism and sun exposure (lower lip), since these conditions were the outcome of some studies included in the review.

Answer: We have included other associated risk factors to oral cancers and potentially malignant disorders in the introduction part. (Line 58 & 76).

Question: Introduction - In the sentence "To the best of our knowledge, there is a systematic review about the oral health of seafarer with underrepresented data about oral health in civilian sector and less developed countries." Perhaps because these primary studies did not exist at the time of this previous review...

Answer: thank you for the insights, it is true that many studies about oral health of fishermen and seafarer in developing countries have published after the period of 2010. We have revised this sentence based on your comment. (Line 95) 

Question: Methods - The authors should adopt the updated version of PRISMA published in 2020. 

Answer: we have updated to the latest version of Prisma flowchart and presented in S1 Figure.

Question: The databases cited in the methodology differ from those cited in the abstract.

Answer: we have changed to the database name to make it consistent with the methodology.

Question: Results - In the sentence "Of the 25 retrieved articles, six articles were excluded because of..." Articles excluded in phase 2 must be cited in the text or the list of the excluded references and justification must be provided in the appendices.

Answer: we thank the reviewer for this suggestion. We have updated the search and included in-text citations of all studies that were excluded in phase two. (Line 146)

Question: Results - In this sentence "In general, 13 selected scoring articles 6 to 9 were considered as having low risk of bias and acceptable quality. While five studies scored 2 to 5, which can be seen as having high risk of bias. When assessing the quality of the RCT, seven out of 11 items are fulfilled, and the quality of the RCT can be rated as moderate risk of bias.” The authors could cite the respective references as it is an important result of RS and needs to be easily accessible to the reader.

Answer: we have added respective references in this part in the S3 Appendix. (Line 196)

Question: Results - In the sentence “The prevalence of smoking among seafarers was 25% and 56%, respectively.” Are these two prevalences from the studies mentioned later or is there any missing information?

Answer: thank you for your feedback, there was some missed data in this part, we have added more detail about prevalence of tobacco consumption (both smoking and smokeless) in this part. (Line 270)

Question: Discussion: It is unclear how the authors evaluated the data from the primary studies of high risk of bias in the analysis of evidence from the frequent oral conditions in the seafarers and fisherman.

Answer: we have evaluated the data from studies that have poor quality in this systematic review under the discussion part. (Line 298).

---

## [Decision Letter · Decision Letter 2]

6 Oct 2023

Overview of Oral Health Status and Associated Risk Factors in Maritime Settings: An Updated Systematic Review

PONE-D-22-13693R2

Dear Dr. Sanju Gautam,

We’re pleased to inform you that your manuscript has been judged scientifically suitable for publication and will be formally accepted for publication once it meets all outstanding technical requirements.

Kind regards,

Tanay Chaubal

Academic Editor

PLOS ONE

Additional Editor Comments (optional):

Reviewers' comments:

Reviewer's Responses to Questions

**Comments to the Author**

1. If the authors have adequately addressed your comments raised in a previous round of review and you feel that this manuscript is now acceptable for publication, you may indicate that here to bypass the “Comments to the Author” section, enter your conflict of interest statement in the “Confidential to Editor” section, and submit your "Accept" recommendation.

Reviewer #1: All comments have been addressed

Reviewer #2: All comments have been addressed

2. Is the manuscript technically sound, and do the data support the conclusions?

Reviewer #1: Yes

Reviewer #2: Yes

3. Has the statistical analysis been performed appropriately and rigorously? 

Reviewer #1: Yes

Reviewer #2: N/A

4. Have the authors made all data underlying the findings in their manuscript fully available?

Reviewer #1: Yes

Reviewer #2: Yes

5. Is the manuscript presented in an intelligible fashion and written in standard English?

Reviewer #1: Yes

Reviewer #2: Yes

6. Review Comments to the Author

Reviewer #1: Thanks for the review conducted. Congratulations to the authors for the work. I have no further questions.

Reviewer #2: The authors have answered sufficiently to the reviewers comments, and the article is deemed suitable for publication.

7. PLOS authors have the option to publish the peer review history of their article (what does this mean?). If published, this will include your full peer review and any attached files.

Reviewer #1: No

Reviewer #2: **Yes: **Camila B Gallo

---

## [Editor Report · Acceptance letter]

10 Oct 2023

PONE-D-22-13693R2 

Overview of oral health status and associated risk factors in maritime settings: an updated systematic review 

Dear Dr. Gautam:

I'm pleased to inform you that your manuscript has been deemed suitable for publication in PLOS ONE. Congratulations! Your manuscript is now with our production department. 

Kind regards, 

on behalf of

Dr. Tanay Chaubal 

Academic Editor

PLOS ONE